# FINE-R1: MAKE MULTI-MODAL LLMS EXCEL IN FINE-GRAINED VISUAL RECOGNITION BY CHAIN-OF-THOUGHT REASONING

**Hulingxiao He, Zijun Geng, Yuxin Peng**[*]
Wangxuan Institute of Computer Technology, Peking University
hehulingxiao@stu.pku.edu.cn, gengzijun2024@163.com, pengyuxin@pku.edu.cn

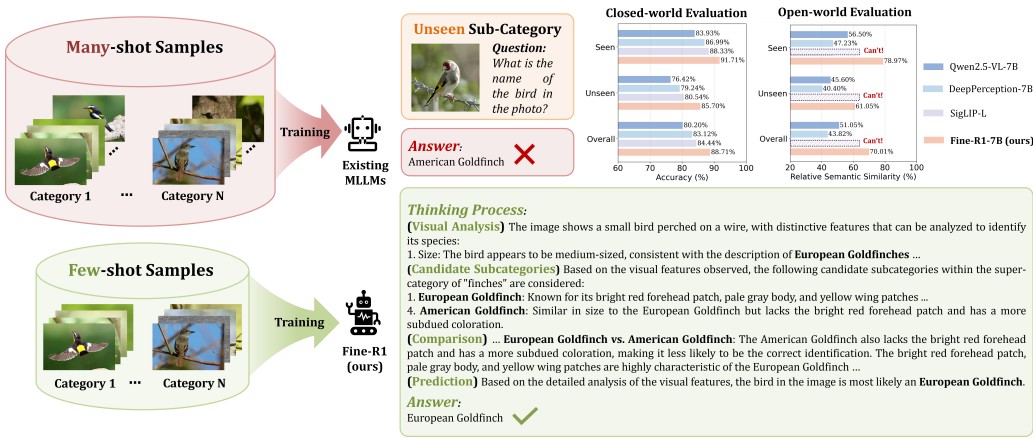

Figure 1: Fine-R1 generates Chain-of-Thought (CoT) before producing the final fine-grained visual recognition (FGVR) answer. It utilizes CoT supervised fine-tuning (SFT) and Triplet Augmented Policy Optimization (TAPO), learning the reasoning process with only few-shot samples per category. In comparison to general and reasoning MLLMs, and contrastive CLIP models, Fine-R1 excels in identifying both seen and unseen categories.

## ABSTRACT

Any entity in the visual world can be hierarchically grouped based on shared characteristics and mapped to fine-grained sub-categories. While Multi-modal Large Language Models (MLLMs) achieve strong performance on coarse-grained visual tasks, they often struggle with Fine-Grained Visual Recognition (FGVR). Adapting general-purpose MLLMs to FGVR typically requires large amounts of annotated data, which is costly to obtain, leaving a substantial performance gap compared to contrastive CLIP models dedicated for discriminative tasks. Moreover, MLLMs tend to overfit to seen sub-categories and generalize poorly to unseen ones. To address these challenges, we propose **Fine-R1**, an MLLM tailored for FGVR through an R1-style training framework: (1) Chain-of-Thought Supervised Fine-tuning, where we construct a high-quality FGVR CoT dataset with rationales of "visual analysis, candidate sub-categories, comparison, and prediction", transition the model into a strong open-world classifier; and (2) Triplet Augmented Policy Optimization, where Intra-class Augmentation mixes trajectories from anchor and positive images within the same category to improve robustness to intra-class variance, while Inter-class Augmentation maximizes the response distinction conditioned on images across sub-categories to enhance discriminative ability. With only 4-shot training, Fine-R1 outperforms existing general MLLMs, reasoning MLLMs, and even contrastive CLIP models in identifying both seen and unseen sub-categories, showing promise in working in knowledge-intensive domains where gathering expert annotations for all sub-categories is arduous. Code is available at https://github.com/PKU-ICST-MIPL/FineR1_ICLR2026.

---

[*]Corresponding author.

# 1 INTRODUCTION

The visual world exhibits inherently fine-grained characteristics, which pose significant challenges for visual understanding. Objects are organized hierarchically according to shared traits and span a vast number of fine-grained categories (Zhang et al., 2024c). For example, the coarse-grained super-category *bird* can be further subdivided into thousands of fine-grained sub-categories such as *Acadian Flycatcher*, *Great Crested Flycatcher*, and *Least Flycatcher*. Moreover, new categories can emerge in real-world applications, requiring to identify unseen concepts (Geng et al., 2020). According to the latest statistics from the International Ornithologists' Union (IOC), as of 2024, 11,276 bird species have been identified worldwide, and the number continues to grow with new species discovered.

Recent advances in Multi-modal Large Language Models (MLLMs) have achieved impressive results on general vision-language tasks such as image captioning and visual question answering (Liu et al., 2024a;b). However, prior studies (Zhang et al., 2024e; Liu et al., 2024c; He et al., 2025) reveal a substantial drop in performance when MLLMs are applied to knowledge-intensive fine-grained visual recognition (FGVR) task. FGVR, a long-standing challenge in computer vision, requires distinguishing subtle differences among visually similar categories, such as animal species (Wah et al., 2011), plant varieties (Nilsback & Zisserman, 2008), or specific models of cars (Krause et al., 2013) and aircraft (Maji et al., 2013). These tasks are difficult even for humans, as they demand extensive domain knowledge to resolve minimal inter-class variance and large intra-class variations. Notably, even state-of-the-art generative MLLMs such as GPT-4 (Achiam et al., 2023) and GeminiPro (Team et al., 2023) underperform compared to contrastive CLIP models (Radford et al., 2021; Zhai et al., 2023) dedicated for discriminative tasks.

To improve FGVR capability, early studies have explored fine-tuning MLLMs with classification data (Zhang et al., 2024e; He et al., 2025; Shi et al., 2025b). While this can yield gains, transforming general-purpose MLLMs into fine-grained classifiers requires extensive labeled data, which is costly to obtain. Moreover, these models tend to overfit to seen categories during training, limiting their utility in real-world scenarios where recognition of novel concepts is crucial (Geng et al., 2020).

To address these challenges, we propose a framework that enables MLLMs to *deploy intrinsic knowledge for FGVR while generalizing effectively to unseen categories with limited data*. It comprises two key components: (1) a two-stage training framework while chain-of-thought supervised fine-tuning (CoT SFT) establishes foundational FGVR capabilities through knowledge-integrated reasoning chains, followed by reinforcement learning that optimizes the capability to deploy knowledge for FGVR via reward signals; and (2) a policy gradient algorithm named Triplet Augmented Policy Optimization (TAPO) designed to tackle the problem of high intra-class variance and low inter-class variance for FGVR. The key idea behind TAPO is to introduce implicit contrastive signals by a positive and negative sample for the anchor image. By mixing trajectories from both anchor and positive image sampled from the same sub-category, it demonstrates improved robustness. By maximize two versions of policy, conditioned on either the anchor or negative image sampled from the most similar sub-category, it encourages the model to distinguish visually-similar objects. Through this framework, we develop **Fine-R1**, an MLLM that enhances FGVR guided by strong CoTs.

Extensive experiments on six FGVR datasets under the few-shot base-to-new generalization setting yield three main findings: (1) State-of-the-art performance: Fine-R1 achieves superior results in both closed-world and open-world settings, surpassing general MLLMs (e.g., closed: +8.51% and open: +23.75% over Qwen2.5-VL-7B), reasoning MLLMs (e.g., closed: +5.59% and open: +30.98% over DeepPerception-7B), and even contrastive CLIP models (e.g., closed: +4.27% over SigLIP-L) dedicated for discriminative tasks. (2) Stronger generalization. Fine-R1-3B exhibits superior cross-domain generalization on unseen categories (e.g., +15.59% over SFT, +10.28% over CLS-RL (Li et al., 2025), and +10.05% over No-Thinking Reinforcement Learning (No-Thinking RL) (Li et al., 2025)), confirming that its improvements stem from enhanced knowledge deployment rather than memorization. (3) Broader benefits. Fine-R1 provides more accurate answers to non-classification questions where object recognition is a prerequisite (e.g., +3.60% over Qwen2.5-VL-3B on ImageWikiQA), while preserving or even surpassing on general VQA tasks.

In summary, our contributions are threefold: (1) We propose Fine-R1, an MLLM with enhanced ability to deploy intrinsic knowledge for FGVR on seen categories while simultaneously demonstrating generalization to unseen categories, merely with few-shot data available. To achieve this,

we develop a two-stage framework integrating CoT SFT with TAPO, dedicated for the problem of high intra-class and low inter-class variances. (2) Extensive experiments demonstrate the strong FGVR capability of Fine-R1 in both closed-world and open-world evaluation, and superior base-to-new category generalization performance. Notably, Fine-R1 surpasses MLLMs with larger parameters and reasoning ability, and even CLIP models dedicated for discriminative tasks. (3) Through a series of analyses, we show that both visual features extracted and knowledge about fine-grained sub-categories do not change substantially through our training, but Fine-R1 does improve the deployment of this knowledge in the context of FGVR task.

## 2 RELATED WORK

**MLLMs for FGVR.** Recent efforts to adapt MLLMs for FGVR either fine-tune them with classification data or design training-free approaches (Peng et al., 2025). Fine-tuning studies show that explicit object mentions in training data are crucial (Geigle et al., 2024; Zhang et al., 2024e), and that integrating sufficient classification-focused data enables MLLMs to bridge the gap between state-of-the-art classifiers while enhancing object-centric reasoning. Some works attribute underperformance to misalignment between visual objects and category names, addressing it through attribute-based alignment (He et al., 2025) or interpretable data synthesis (Shi et al., 2025a). In contrast, training-free methods such as Sparse Attention Vectors exploit sparse attention activations for discriminative tasks (Mitra et al., 2024a), but performance remains limited without large annotated datasets.

**Reinforcement Learning.** Reinforcement learning (RL) has shown strong potential in enhancing reasoning abilities of both LLMs and MLLMs by reducing reliance on large annotated datasets (Yue et al., 2024). In LLMs, the GPT series (Hurst et al., 2024; Achiam et al., 2023; Jaech et al., 2024) leveraged RL with human feedback (Ouyang et al., 2022) to optimize reasoning, excelling in mathematics (Cai et al., 2024; Ying et al., 2024; Shao et al., 2024; Yang et al., 2024; Luong et al., 2024) and coding tasks (Hui et al., 2024; Zhang et al., 2024a;f). DeepSeek-R1-Zero (Guo et al., 2025) further demonstrated RL-only training for reasoning improvement. In MLLMs, a series of work (Liu et al., 2025b; Li et al., 2025; Huang et al., 2025; Ma et al., 2025) advanced visual perception and inference through RL with verifiable rewards. Together, these works highlight RL's effectiveness in interactive visual-linguistic reasoning. While policy optimization has shown great potential in image classification task (Liu et al., 2025b; Ma et al., 2025) for MLLMs, how it can be optimized for solving the key challenge of fine-grained image classification task hasn't been investigated. Instead, we equip policy optimization with contrastive paradigms to make the model more robust to the high intra-class variance and discriminative to the low inter-class variance.

**CoT Reasoning with MLLMs.** Chain-of-thought (CoT) reasoning provides explicit intermediate steps that connect a problem to its solution (Wei et al., 2022b), and such rationales have been shown to substantially improve LLM reasoning (Cheng et al., 2024; Fu et al., 2023b; Wang et al., 2023; Diao et al., 2023). In multimodal contexts, CoT reasoning is enabled through two complementary strategies: CoT prompting (Gao et al., 2024; Mitra et al., 2024b; Lu et al., 2023) and CoT SFT (Luo et al., 2025a; Xu et al., 2024; Thawakar et al., 2025). These approaches empower MLLMs to tackle challenging tasks including visual math reasoning (Zhang et al., 2024b) and embodied decision-making (Mu et al., 2023). CoT prompting is typically applied in zero-shot (Kojima et al., 2022) or few-shot (Zhang et al., 2023) settings, guiding models such as GPT-4o (Hurst et al., 2024) and Claude 3.5 Sonnet (Anthropic, 2024) to articulate reasoning steps before answering. In contrast, CoT SFT relies on multimodal instruction tuning with datasets containing explicit reasoning traces, and its success hinges on the quality of these examples. In this work, we adopt CoT SFT with AI-generated and human-verified samples. Rather than using generic "visual analysis and prediction" rationales, we utilize the structured reasoning procedure of "visual analysis, candidate subcategories, comparison, and final prediction", eliciting the model to first propose candidate subcategories (the most likely categories base model confuses it for) and then utilize text knowledge to resolve this confusion by detailed comparison between candidates.

## 3 PRELIMINARIES

**FGVR with MLLMs.** Let us define an MLLM as a function $f_{\mathrm{MLLM}}$ generating a text output $y$ in the space $\mathcal{T}$ given an image $x$ in the space $\mathcal{X}$ and a text query $q \in \mathcal{T}$, $f_{\mathrm{MLLM}} : \mathcal{X} \times \mathcal{T} \to \mathcal{T}$. To

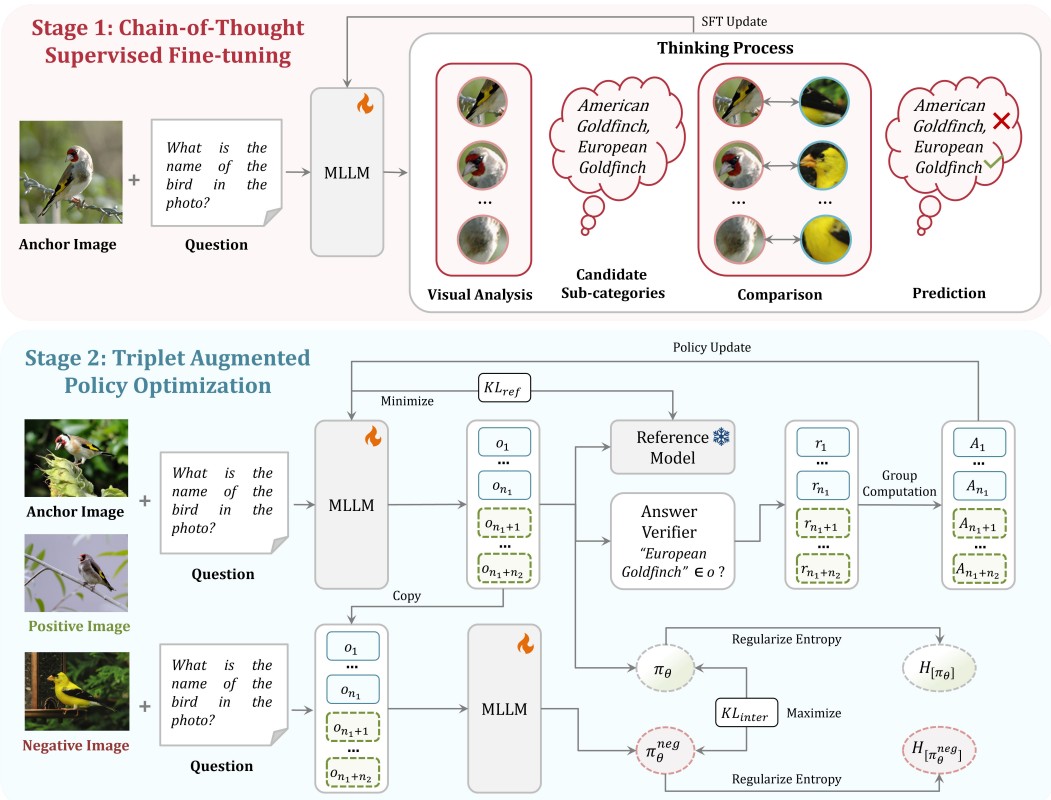

Figure 2: Overview of the proposed two-stage training framework integrating CoT SFT and TAPO.

perform FGVR with MLLMs in the open-world setting, the query $q$ contains a prompt of the type *"What is the name of the bird in the photo?"*. We let MLLM predict naturally on its original output space $\mathcal{T}$ without any constraint, and expect the output $y$ to be a sub-category $c \in \mathcal{T}$. In the case of closed-world setting, we have a predefined list $\mathcal{C}$ of classes and we modify $q$ by specifying the set $\mathcal{C}$ via a multi-choice question. As a consequence, the model is required to pick from the set $\mathcal{C}$ of all candidate subcategories, with $c \in \mathcal{C}$.

**FGVR with CLIP models.** Let us define a CLIP model as two mapping functions: $f_{\text{text}}$ generating text embedding $e_{\text{text}}$ in the space $\mathcal{E}$ given a text input $t$, $f_{\text{text}} : \mathcal{T} \to \mathcal{E}$; $f_{\text{image}}$ generating image embedding $e_{\text{image}}$ in the space $\mathcal{E}$ given an image $x$, $f_{\text{image}} : \mathcal{X} \to \mathcal{E}$. CLIP models can only be used in a closed-world setting where the label set $\mathcal{C}$ is known. Following (Radford et al., 2021), we use the prompt "a photo of a <class>" as the text input $t$. The sub-category with the highest cosine similarity to the image embedding $e_{\text{image}}$ is selected as the predicted answer: $c = argmax_{i \in C} \ Sim(e^i_{\text{text}}, e_{\text{image}})$.

## 4 METHODOLOGY

### 4.1 OVERVIEW

In this section, we introduce our Fine-R1 model and the associated progressive two-stage framework. As shown in Figure 2, this method begins with chain-of-thought supervised fine-tuning (CoT SFT), which teaches the model to perform open-world FGVR in a "human-like" manner by SFT. Then, we apply our proposed Triplet Augmented Policy Optimization (TAPO) to guide the model to explore potentially better thinking process that is robust to intra-class variance and discriminative with respect to inter-class variance.

## 4.2 Chain-of-Thought Supervised Fine-tuning

The primary objective of the first stage training is to enhance models' open-world FGVR capabilities by training them on synthesized CoT reasoning data, as illustrated in Figure 2. This process endows the model with a structured reasoning procedure of "*visual analysis, candidate subcategories, comparison, and final prediction*" by explicitly imitating human-like reasoning pathways. Such structured training lays the foundation for forming knowledge-association patterns that support subsequent reinforcement learning (RL) optimization.

**Open-world CoT Data.** For data construction, we sample one image per sub-category and employ Qwen2.5-VL-32B (Bai et al., 2025), to generate open-world FGVR CoT data in two key stages: (1) Image-level Visual Concept Selection (Shi et al., 2025b): Given an image and its ground-truth subcategory, we first extract image-specific concepts that capture the connection between visual content and the subcategory. Specifically, we leverage the MLLM's captioning ability to produce multiple descriptions of the same image, each emphasizing different visual attributes. By aggregating this diverse set of descriptions, we approximate the distribution of discriminative features and mitigate the incompleteness of individual captions. To refine these features, we further apply an information bottleneck strategy, retaining only the most relevant ones. This process explicitly transforms critical visual cues into textual representations, providing a richer foundation for reasoning than raw image inputs alone. See qualitative examples of the extracted visual concepts associated with each image in Appendix A. (2) Structured CoT Prompt: The extracted visual concepts are concatenated with the image-question pair to guide the MLLM in focusing on discriminative details for FGVR. Inspired by human cognition, we design a structured CoT prompt that decomposes the reasoning process into clear stages: visual analysis, candidate subcategories, comparison, and final prediction. An example of the CoT prompt template is shown in Appendix B. To ensure data reliability, we design some dedicated strategies, ultimately yielding a high-quality open-world FGVR CoT dataset containing 404 samples: (1) Multiple responses are sampled until the CoT leads to exactly matched subcategory. (2) We detect CoT with mixed language and manually correct them to English. (3) We manually check the predicted subcategory in the CoT rationales, and maintain the samples whose predictions are both included in the candidate subcategories and consistent with the ground truth.

**CoT SFT.** We fine-tune the model on the curated CoT dataset, enabling it to develop strong open-world FGVR capabilities. Through this process, the model learns to integrate domain knowledge when generating candidate subcategories and to conduct meticulous comparisons that lead to more accurate predictions. The resulting model provides a robust foundation for further optimization through RL in the subsequent training stage.

## 4.3 Triplet Augmented Policy Optimization

Following CoT SFT, we further explore the use of Decoupled Clip and Dynamic Sampling Policy Optimization (DAPO) (Yu et al., 2025) to enhance the model's open-world FGVR capabilities, building upon the structured reasoning foundation. DAPO, a representative successor to GRPO (Shao et al., 2024), introduces several improvements such as Clip-Higher, Dynamic Sampling, and Token-Level Policy Gradient Loss. To address the unique challenges of FGVR, we propose **Triplet Augmented Policy Optimization (TAPO)**. The central idea is to encourage the policy to remain discriminative under low inter-class variance while maintaining robustness under high intra-class variance when predicting the final answer. Specifically, we augment each anchor image $x$ with a positive image $x_{\text{pos}}$ from the same subcategory and a negative image $x_{\text{neg}}$ from the most visually similar but distinct subcategory. This forms triplets $T = (x, x_{\text{pos}}, x_{\text{neg}})$, supporting both intra-class and inter-class augmentation.

**Intra-class Augmentation.** To improve robustness against substantial intra-class variation, we introduce Intra-class Augmentation, a strategy designed to enrich sample diversity within target classes. Inspired by (Liu et al., 2025a), IntraClassAug employs a hybrid sampling approach that leverages predicted answers from both anchor images $x$ and their positive counterparts $x_{\text{pos}}$. This approach captures a broader range of intra-class variants. Concretely, for each input pair $(x, q)$, we randomly sample a positive image $x_{\text{pos}}$ from the same category. As illustrated in Figure 2, the old policy $\pi_{\theta_{\text{old}}}$ generates two sets of rollouts: $n_1$ responses conditioned on the anchor $(x, q)$ and $n_2$ responses conditioned on the positive $(x_{\text{pos}}, q)$. All rollouts are then aggregated into a single pool

for reward computation:

$$\mathbf{r} = \{r_i\}_{i=1}^{n_1+n_2} = \{r(x, q, o_j)\}_{j=1}^{n_1} \cup \{r(x_{\text{pos}}, q, o_k)\}_{k=n_1+1}^{n_1+n_2}. \tag{1}$$

Importantly, the policy update step remains conditioned only on the anchor $(x, q)$, promoting more effective exploration.

This design yields two main advantages for FGVR: (1) Intra-class diversity modeling: Positive trajectories from different images within the same subcategory introduce varied visual perspectives, helping the policy better capture subtle within-class variations. (2) Discriminative guidance: When anchor and positive images yield different predictions for the same query, discrepancies in rewards provide informative signals that encourage the model to focus on fine-grained, image-specific cues, thereby strengthening category-level distinctions.

**Inter-class Augmentation.** To address the challenge of low inter-class variance, we propose Inter-class Augmentation, which encourages the policy to generate distinct responses for visually similar images from different subcategories. To quantify whether a model can effectively distinguish between matched and mismatched image–category pairs, we define the ratio:

$$g^{\text{inter}}(\theta) = \frac{\pi_\theta(o \mid q, x_*)}{\pi_\theta(o \mid q, x_{\text{neg}})} \tag{2}$$

where $o$ is a generated sequence of tokens, $q$ is the question and $x_*$ is the anchor image $x$ or positive image $x_{\text{pos}}$. This ratio measures how much the model's output distribution changes when the input image is replaced by a near-neighbor from another sub-category. A higher ratio indicates that the model assigns significantly lower probability to the correct output under the negative image, suggesting that it has learned to leverage category-specific discriminative features. Conversely, a low ratio suggests that predictions remain largely unchanged, even when informative features are removed and misleading ones introduced, implying difficulty in localizing fine-grained cues. Thus, for a well-trained model $\theta$, we expect $g^{\text{inter}}(\theta)$ to be high.

Inspired by (Wang et al., 2025), we introduce an additional loss term into the DAPO objective by maximizing the KL divergence between the output distribution conditioned on the anchor/positive image and that conditioned on the negative image:

$$\mathbb{D}_{\text{KL}}[\pi_\theta || \pi_\theta^{\text{neg}}] = \mathbb{D}_{\text{KL}}[\pi_\theta(o|q, x_*) \parallel \pi_\theta(o|q, x_{\text{neg}})] \tag{3}$$

Combining intra-class and inter-class augmentation, the resulting objective is:

$$\mathcal{J}_{\text{TAPO}}(\theta) = \mathbb{E}_{[(x,q)\sim p_\mathcal{D}, \{o_j\}_{j=1}^{n_1}\sim\pi_{\theta_{\text{old}}}(\cdot|q,x), \{o_k\}_{k=n_1+1}^{n_1+n_2}\sim\pi_{\theta_{\text{old}}}(\cdot|q,x_{\text{pos}})]} \frac{1}{\sum_{i=1}^{n_1+n_2}|o_i|} \sum_{i=1}^{n_1+n_2} \sum_{t=1}^{|o_i|} \Big\{$$

$$\min\left[r_{i,t}(\theta)\hat{A}_{i,t}, \text{clip}\left(r_{i,t}(\theta), 1-\epsilon_l, 1+\epsilon_h\right)\hat{A}_{i,t}\right] + \gamma\mathbb{D}_{\text{KL}}[\pi_\theta||\pi_\theta^{\text{neg}}] - \eta_1\mathcal{H}[\pi_\theta] - \eta_2\mathcal{H}[\pi_\theta^{\text{neg}}]\Big\}$$

$$\text{with } 0 < |\{o_i \mid \text{is\_included}(a, o_i)\}| < n_1 + n_2 \tag{4}$$

where $\gamma$ is the weight for the KL-divergence term $\mathbb{D}_{\text{KL}}[\pi_\theta||\pi_\theta^{\text{neg}}] = g_i^{\text{inter}}(\theta) - \log g_i^{\text{inter}}(\theta) - 1$ following (Hershey & Olsen, 2007). Here, $i$ indexes the $i$-th rollout response. Since the KL divergence is unbounded, we adopt the double entropy regularization strategy from (Wang et al., 2025) to constrain both $\pi_\theta$ and $\pi_\theta^{\text{neg}}$, thereby encouraging stable and low-entropy distributions. The entropies are defined as $\mathcal{H}[\pi_\theta] = \log\pi_\theta(o|q, x_*)$, $\mathcal{H}[\pi_\theta^{neg}] = \log\pi_\theta(o|q, x_{\text{neg}})$. $\text{is\_included}(a, o_i)$ checks whether the ground truth is contained in the response. $\eta_1$ and $\eta_2$ are hyperparameters used to weight the corresponding loss terms.

## 5 EXPERIMENTS

### 5.1 EXPERIMENT SETTINGS

**Datasets.** We conduct experiments on several popular FGVR datasets that include CaltechUCSD Bird-200 (Wah et al., 2011), Stanford Car-196 (Krause et al., 2013), Stanford Dog-120 (Krause et al.,

Table 1: Closed-world FGVR evaluations in terms of accuracy (%). The best results are **bolded** and the second best results are underlined in all following tables. All results are averaged with 3 trials.

| Models | Seen Categories | | | | | | | Unseen Categories | | | | | | | Avg. |
|---|---|---|---|---|---|---|---|---|---|---|---|---|---|---|---|
| | Air. | Bird | Car | Dog | Flower | Pet | Avg. | Air. | Bird | Car | Dog | Flower | Pet | Avg. | |
| **CLIP Models** | | | | | | | | | | | | | | | |
| CLIP-L | 47.95 | 73.96 | 80.50 | 77.12 | 87.59 | 95.45 | 77.10 | 45.68 | 62.35 | 79.81 | 73.58 | 57.24 | 89.22 | 67.98 | 72.54 |
| EVA-G | 40.96 | 81.37 | 90.06 | 76.02 | 81.70 | 94.21 | 77.39 | 45.83 | 63.39 | 87.06 | 76.66 | 54.83 | 89.82 | 69.60 | 73.49 |
| SigLIP-L | 67.08 | 85.10 | **96.03** | 86.18 | **97.59** | **98.02** | 88.33 | 69.35 | 74.17 | **92.97** | 84.44 | 69.07 | 93.24 | 80.54 | 84.44 |
| SigLIP2-L | 43.36 | 71.26 | 93.43 | 78.51 | 89.05 | 92.14 | 77.96 | 40.87 | 58.03 | 90.00 | 77.15 | 61.39 | 93.03 | 70.08 | 74.02 |
| **General MLLMs** | | | | | | | | | | | | | | | |
| Idefics2-8B | 49.90 | 52.37 | 90.85 | 58.24 | 82.10 | 81.85 | 69.22 | 48.53 | 40.20 | 84.67 | 44.79 | 60.54 | 83.86 | 60.43 | 64.83 |
| Idefics3-LLaMA3-8B | 43.31 | 34.51 | 75.18 | 47.50 | 65.12 | 72.98 | 56.43 | 48.38 | 37.47 | 70.59 | 43.80 | 55.30 | 68.59 | 54.02 | 55.23 |
| LLaVA-v1.6-mistral-7B | 49.60 | 55.96 | 71.02 | 45.34 | 62.86 | 62.59 | 57.90 | 47.71 | 50.32 | 67.62 | 37.20 | 46.16 | 65.71 | 52.45 | 55.17 |
| LLaVA-Onevision-7B | 32.07 | 54.81 | 71.38 | 70.72 | 73.83 | 76.52 | 63.22 | 30.43 | 52.92 | 67.21 | 53.78 | 48.94 | 66.31 | 53.27 | 58.24 |
| InternVL2.5-2B | 36.71 | 65.23 | 63.40 | 53.70 | 65.39 | 75.46 | 59.98 | 40.57 | 62.74 | 63.53 | 42.62 | 46.35 | 60.21 | 52.67 | 56.33 |
| InternVL2.5-4B | 38.31 | 29.54 | 51.44 | 34.65 | 61.44 | 51.01 | 44.40 | 39.29 | 31.29 | 44.95 | 31.99 | 35.83 | 54.25 | 39.60 | 42.00 |
| InternVL2.5-8B | 46.70 | 53.14 | 62.17 | 54.26 | 68.25 | 76.65 | 60.20 | 45.38 | 47.12 | 61.15 | 48.82 | 47.95 | 62.83 | 52.21 | 56.20 |
| Qwen2-VL-2B | 66.18 | 60.15 | 94.28 | 64.67 | 91.93 | 85.39 | 77.10 | 65.51 | 44.92 | 87.89 | 58.60 | 50.21 | 79.84 | 64.50 | 70.80 |
| Qwen2-VL-7B | 78.27 | 67.41 | 94.60 | 71.70 | 93.84 | 91.04 | 82.81 | 79.86 | 56.25 | 89.63 | 66.16 | 67.47 | 78.30 | 72.95 | 77.88 |
| Qwen2.5-VL-3B | 64.24 | 65.40 | 86.70 | 70.51 | 94.24 | 83.46 | 77.43 | 68.29 | 58.98 | 80.65 | 67.10 | 68.74 | 87.88 | 71.94 | 74.68 |
| Qwen2.5-VL-7B | 74.28 | 70.54 | 90.75 | 80.19 | 96.20 | 91.91 | 83.98 | 71.60 | 66.29 | 84.02 | 77.54 | 65.63 | 93.44 | 76.42 | 80.20 |
| **Reasoning MLLMs** | | | | | | | | | | | | | | | |
| DeepPerception-7B | **83.52** | 74.16 | 94.89 | 80.40 | 97.05 | 91.91 | 86.99 | **86.48** | 61.19 | 89.72 | 77.85 | 72.80 | 87.41 | 79.24 | 83.12 |
| **Fine-R1-3B (ours)** | 76.87 | 86.79 | 92.14 | 87.85 | 96.25 | 93.89 | 88.97 | 75.73 | 79.10 | 87.40 | 80.93 | 73.93 | 91.36 | 81.41 | 85.19 |
| **Fine-R1-7B (ours)** | 82.32 | **90.50** | 94.03 | **90.11** | 97.22 | 96.05 | **91.71** | 77.91 | **87.54** | 87.99 | **89.71** | 74.12 | **96.92** | **85.70** | **88.71** |

2013), Flower-102 (Nilsback & Zisserman, 2008), Oxford-IIIT Pet-37 (Parkhi et al., 2012), and FGVC-Aircraft (Maji et al., 2013). To facilitate evaluation on base-to-new category generalization, we randomly select 60% of the categories as seen categories and the remaining 40% as unseen categories for each dataset. We train a unified model for all six datasets with 4-shot data per seen category, and do evaluation on test sets of the seen and unseen categories, respectively.

**Evaluation Metrics.** We define success on a single example as whether the ground-truth choice is included in the MLLM generation. We report the success rate of all test examples as the accuracy in the closed-world setting. Since evaluating models in the open-world setting presents additional challenges, as predictions may differ in granularity (e.g., Boeing 737 vs. Boeing 737-200), or ground truth may include redundancy for distinguishing from others (e.g.,"Coupe 2012" in Audi A5 Coupe 2012 and Audi S5 Coupe 2012), we use two complementary metrics: (1) text inclusion (Zhang et al., 2024e), evaluating strict string matching. (2) relative semantic similarity between the text embeddings of predictions and ground truth calculated by the SigLIP (Zhai et al., 2023) text encoder. Instead of using the similarity as reward directly, we use the similarity between the super-category and the ground truth subcategory as the standard to calculate the relative value. Formally, the relative semantic similarity $SS_{\text{relative}}$ is expressed as:

$$SS_{\text{relative}} = \ max(0, \frac{Sim(c,c^*)-Sim(\hat{c},c^*)}{1-Sim(\hat{c},c^*)}), \tag{5}$$

where $\hat{c}$, $c$, and $c^*$ denote the super-category, predicted and ground truth subcategory, respectively. We defer prompts for evaluations, implementation details, and compared models to Appendix B, C, and D, respectively.

## 5.2 MAIN RESULTS

**Closed-world Evaluation.** As shown in Table 1, although trained solely on open-world FGVR tasks, Fine-R1 achieves substantial performance gains in the closed-world setting with the guidance of CoT. This validates our hypothesis that a human-inspired reasoning process enables Fine-R1 to better distinguish visually similar sub-categories. On seen categories, Fine-R1-7B reaches an ac-

Table 2: Open-world FGVR evaluations in terms of relative semantic similarity (%). All results are averaged with 3 trials.

| Models | Seen Categories | | | | | | | Unseen Categories | | | | | | | Avg. |
|---|---|---|---|---|---|---|---|---|---|---|---|---|---|---|---|
| | Air. | Bird | Car | Dog | Flower | Pet | Avg. | Air. | Bird | Car | Dog | Flower | Pet | Avg. | |
| **General MLLMs** | | | | | | | | | | | | | | | |
| Idefics2-8B | 3.64 | 19.68 | 19.54 | 10.03 | 14.94 | 2.50 | 11.72 | 3.69 | 15.35 | 20.81 | 10.19 | 5.84 | 2.47 | 9.73 | 10.72 |
| Idefics3-LLaMA3-8B | 9.66 | 27.72 | 22.96 | 35.39 | 40.92 | 20.17 | 26.14 | 7.09 | 24.93 | 22.08 | 27.67 | 21.99 | 24.84 | 21.43 | 23.79 |
| LLaVA-v1.6-mistral-7B | 2.73 | 16.02 | 21.75 | 19.35 | 10.33 | 12.47 | 13.78 | 2.51 | 17.40 | 23.24 | 18.16 | 8.20 | 10.62 | 13.36 | 13.57 |
| LLaVA-Onevision-7B | 9.90 | 31.74 | 21.35 | 30.13 | 45.24 | 17.87 | 26.04 | 7.47 | 29.81 | 19.56 | 25.44 | 16.53 | 19.35 | 19.69 | 22.87 |
| InternVL2.5-2B | 7.26 | 21.32 | 27.29 | 25.87 | 23.08 | 26.56 | 21.90 | 5.22 | 20.12 | 24.73 | 24.84 | 12.59 | 24.55 | 18.68 | 20.29 |
| InternVL2.5-4B | 14.71 | 25.41 | 32.19 | 33.13 | 23.84 | 28.19 | 26.25 | 12.64 | 23.91 | 29.11 | 30.80 | 13.10 | 26.32 | 22.65 | 24.45 |
| InternVL2.5-8B | 23.76 | 28.44 | 30.08 | 27.11 | 21.73 | 29.03 | 26.69 | 20.34 | 24.38 | 27.55 | 24.73 | 13.56 | 26.23 | 22.80 | 24.75 |
| Qwen2-VL-2B | 47.49 | 48.72 | 52.95 | 51.22 | 66.56 | 19.88 | 47.80 | 48.88 | 39.32 | 49.33 | 45.16 | 33.87 | 22.43 | 39.83 | 43.82 |
| Qwen2-VL-7B | 56.47 | 56.46 | 55.31 | 67.03 | 75.02 | 36.97 | 57.88 | 52.75 | 41.17 | 52.46 | 61.01 | 32.57 | 30.74 | 45.12 | 51.50 |
| Qwen2.5-VL-3B | 56.98 | 66.77 | 52.49 | 65.12 | 68.96 | 26.27 | 56.10 | 52.50 | 48.09 | 51.75 | 59.02 | 34.19 | 28.78 | 45.72 | 50.91 |
| Qwen2.5-VL-7B | 58.86 | 65.97 | 56.94 | 59.02 | 62.61 | 35.59 | 56.50 | 48.62 | 45.26 | 55.39 | 54.59 | 32.74 | 36.98 | 45.60 | 51.05 |
| **Reasoning MLLMs** | | | | | | | | | | | | | | | |
| DeepPerception-7B | 44.24 | 47.63 | 54.14 | 49.16 | 47.30 | 40.90 | 47.23 | 40.03 | 37.10 | 52.27 | 49.05 | 28.38 | 35.57 | 40.40 | 43.82 |
| **Fine-R1-3B (ours)** | 54.36 | 78.90 | 82.46 | 78.21 | 64.55 | 81.60 | 73.35 | 46.43 | 58.00 | 74.60 | 70.08 | 39.54 | 79.11 | 61.29 | 67.32 |
| **Fine-R1-7B (ours)** | 73.53 | 86.12 | 90.73 | 80.71 | 81.46 | 83.14 | 82.62 | 65.21 | 60.69 | 82.19 | 70.97 | 40.74 | 82.04 | 66.97 | 74.80 |

curacy of 91.71%, outperforming all baselines of comparable scale (e.g., +7.73% over Qwen2.5-VL-7B) and even surpassing strong contrastive CLIP models (e.g., +3.38% over SigLIP-L). For unseen categories, it achieves 85.70% accuracy, yielding even larger improvements (e.g., +9.28% over Qwen2.5-VL-7B and +5.16% over SigLIP-L). These results further demonstrate that Fine-R1 not only leverages knowledge effectively for FGVR but also generalizes well to novel categories. Performance comparison in terms of text inclusion is presented in Appendix E.

**Open-world Evaluation.** As shown in Table 2, Fine-R1-7B establishes new state-of-the-art performance with only 4-shot training samples per sub-category, achieving 74.80% relative semantic similarity on average. This represents a substantial improvement of 23.75% over Qwen2.5-VL-7B. Notably, Fine-R1 still demonstrates strong base-to-new category generalization in the open-world setting. The superior performance in both closed-world and open-world FGVR scenarios demonstrates that CoT guidance provides two key advantages: (1) It enhances the model's ability to discern subtle discriminative features among visually similar candidates, and (2) it enables more effective integration of inherent knowledge to identify candidates that accurately capture the ground truth sub-category. A more detailed analysis of the performance gain is presented in Section 5.4.

### 5.3 ABLATION STUDY

We conduct several ablation studies to verify the effectiveness of our design. For the ablation study, we use Qwen2.5-VL-3B and Fine-R1-3B by default.

**Training Methods.** Figure 3a compares different training methods in closed-world setting. SFT greatly improves accuracy on seen categories (+3.98%) but severely harms unseen categories (-6.12%), showing overfitting and poor generalization. CLS-RL (Li et al., 2025) alone reduces the unseen drop but still underperforms the zero-shot baseline (71.13% vs. 71.94%). Moreover, it degrades the accuracy by 5.92% on seen categories as models with limited capabilities struggle to generate high-quality CoT for RL. Though No-Thinking-RL (Li et al., 2025) achieves performance gains on seen categories, it still lags behind SFT (80.86% vs. 81.41%). Our two-stage framework combines the strengths of SFT and RL, significantly outperforming SFT on seen categories (+7.56%) while significantly surpassing No-Thinking-RL on unseen categories (+10.05%).

**Inference Strategies.** We investigate two inference strategies, including CoT prompting, and In-Context Learning (ICL) in the closed-world evaluation. For CoT prompting, we leverage the zero-shot CoT prompting technique by adding "let's think step by step" at the end of the prompt (Wei et al., 2022a; Kojima et al., 2022). For CLIP-like models, we additionally add prompt ensem-

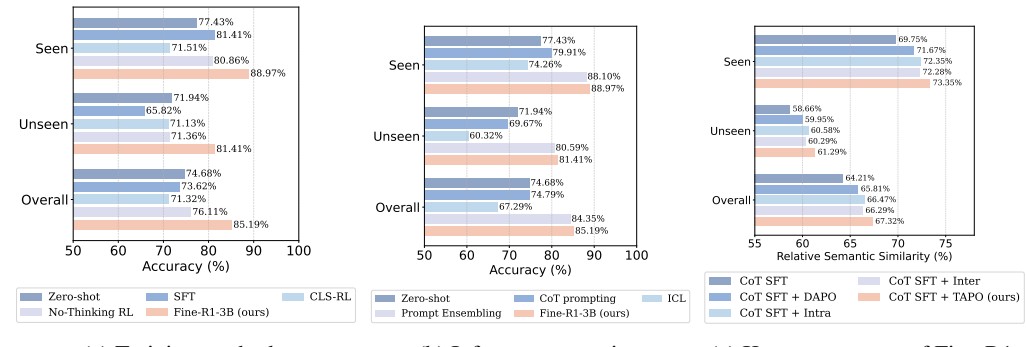

(a) Training methods.     (b) Inference strategies.     (c) Key components of Fine-R1.

Figure 3: Ablation study on training methods, inference strategies, and key components of Fine-R1.

bling results for SigLIP using 80 prompt templates from ImageNet dataset. As shown in Figure 3b, compared to the baseline (74.68%), direct CoT prompting without training for CoT reasoning (74.79%) has a limited impact on FGVR, which is also affirmed in (Zhang et al., 2024d). For ICL, we randomly sample one demonstration for each candidate once it belongs to seen categories (i.e., occurs in the training data). However, since we can only retrieve demonstrations for seen categories in the candidates, the context may introduce bias. The results show that Fine-R1-3B surpasses Qwen2.5-VL-3B with ICL by 17.90%, strengthen the effectiveness of Fine-R1. It is worth noting that Fine-R1 outperforms CLIP-like models even with prompt ensembling optimization.

**Key Components.** As illustrated in Figure 3c, we evaluate the effectiveness of each key component of the training framework. CoT SFT improves relative semantic similarity by 13.30%, laying the foundation for high-quality reasoning. Utilizing DAPO after CoT SFT brings a performance gain of 1.60%, demonstrating effective integration of domain knowledge. Adding Intra-class and Inter-class Augmentation individually both outperform CoT SFT + DAPO, and combining them together achieves the best results of 67.32%, confirming that Fine-R1 benefits from complementary augmentations.

**Anchor-to-Positive Ratio.** As illustrated in Table 3a, we control $n_1 + n_2 = 10$, and change the anchor-to-positive rollout ratio $n_1 : n_2$. Since $n_1 : n_2 = 1$ achieves the best performance, confirming that the performance gain of intra-class augmentation is from increasing the diversity of rollouts instead of generating more rollouts (i.e., using $x$ merely to generate more rollouts).

**CoT Generalization.** We construct more CoT data from the same model Qwen2.5-VL-32B, and evaluate $SS_{relative}$ on unseen categories to test scaling behavior when using larger CoT data. As shown in Table 3b, the model performance increases with the number of CoT data, confirming that the model does not overfit to synthetic patterns in the limited synthetic set. Additionally, we can observe that quality out-weights quantity of CoT data, alleviating the cost to construct a large scale of data for CoT SFT, showing the high data efficiency.

**Cross-model Evaluation.** We conduct experiments on the Qwen2-VL-2B-Instruct (Wang et al., 2024) model to further enhance the evidence of the effectiveness. Architecturally, Qwen2.5-VL differs from Qwen2-VL through the use of an updated vision encoder and language model and a new vision–language fusion module. As shown in Table3c, the consistent gains compared to CoT SFT+DAPO again shows the generality of TAPO to different model architectures.

We defer the general capability analysis and qualitative examples to Appendix G and H.

## 5.4 PERFORMANCE GAIN ANALYSIS

To better understand why Fine-R1 outperforms baselines on FGVR, we propose three hypotheses inspired by the essential capabilities of MLLMs for fine-grained recognition (He et al., 2025). **H1:** Fine-R1 improves the extraction of visual cues needed to distinguish objects; **H2:** Fine-R1 fundamentally enhances knowledge of subcategories; **H3:** Fine-R1 improves the ability to *deploy* existing subcategory knowledge in FGVR tasks. We analyze each hypothesis below.

Table 3: Ablation study on $n_1$:$n_2$, #CoTs in stage 1, and cross-model evaluation.

Table 4: **Left**: Linear probing of visual features and differences. **Right**: Differences in cosine similarities between species pairs belonging to the same and different genus.

(a) $n_1$:$n_2$.

| $n_1$ | $n_2$ | Avg. |
|---|---|---|
| 10 | 0 | 59.47 |
| 8 | 2 | 59.21 |
| **5** | **5** | **61.29** |
| 2 | 8 | 59.33 |
| 0 | 10 | 59.39 |

(b) #CoTs.

| #CoTs | Avg. |
|---|---|
| 404 | 58.66 |
| 804 | 62.06 |
| **1199** | **62.64** |

(c) Cross-model evaluation on Qwen2-VL-2B.

| Method | Avg. |
|---|---|
| Zero-shot | 48.84 |
| CoT SFT + DAPO | 62.32 |
| **CoT SFT + TAPO** | **64.65** |

| Models | Probing (%) | Embedding Similarity | | |
|---|---|---|---|---|
| | | $\Delta$ | $t$ | $p$ |
| Qwen2.5-VL-3B | 84.26 | 0.0088 | | |
| **Fine-R1 (ours)** | 85.00 | 0.0531 | -0.3872 | 0.6993 |

**H1: Fine-R1 extracts better visual cues.** We perform linear probing on image features. Specifically, we retrieve image token embeddings from the residual stream of the final LLM layer, apply mean pooling, and train a linear classifier on CUB-200 training set with batch size 512, learning rate 1e-4, Adam optimizer, and 500 training epochs. The best test performance during training is reported. Results in Table 4 show negligible differences between Fine-R1 and the base model, indicating that Fine-R1 *does not produce more effective visual embeddings for FGVR*.

**H2: Fine-R1 encodes more subcategory knowledge.** We evaluate whether Fine-R1 reserves more knowledge about sub-categories, like taxonomy-aware relationships among bird species. For each target species, we compute similarities with one species from the same genus and with four from different genus, then take the difference between intra-genus and inter-genus similarities. If this difference is larger for Fine-R1, it would suggest stronger taxonomy-aware encoding. However, Table 4 shows little difference between Fine-R1 and Qwen2.5-VL-3B, suggesting that Fine-R1 *does not fundamentally alter subcategory knowledge*.

**H3: Fine-R1 better deploys subcategory knowledge.** We study the distinguishability of positive image–category pairs from negative ones, and examine whether this distinction is reflected in the holistic representation of the input context (e.g., "<image> Is the bird species {correct/incorrect name}?"). Specifically, we use the last hidden state of the final LLM layer as the summary representation of the full context, encompassing both the image and question. We then test whether inputs containing positive pairs can be separated from those containing negative pairs through PCA. Figure 4 presents the first

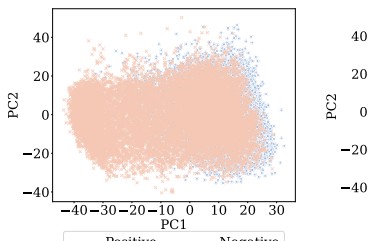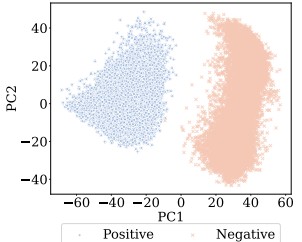

Figure 4: PCA projections of the last hidden state representations of inputs containing positive and negative image-category pairs, extracted from Qwen2.5-VL-3B and Fine-R1.

two principal components of input representations from Qwen2.5-VL-3B and Fine-R1, with positive and negative pairs color-coded. The results show that positive and negative pairs are more linearly separable in Fine-R1 representations, suggesting that Fine-R1 *are better at deploying fine-grained subcategory knowledge,* achieving genuinely different representational states compared to Qwen2.5-VL-3B when the task context requires utilizing knowledge for FGVR.

## 6 CONCLUSION

In this work, we tackle the challenges of data inefficiency and base-to-new generalization in FGVR tasks by proposing a framework that strengthens the ability to leverage intrinsic knowledge through CoT SFT and TAPO. By augmenting policy optimization with triplets consisting of an anchor image, a positive image, and a negative image drawn from the same or different subcategories, our method effectively addresses the issues of high intra-class variance and low inter-class variance. By guiding MLLMs to generate CoTs in a "human-like" manner, Fine-R1 achieves state-of-the-art results in both closed-world and open-world evaluations, outperforming contrastive CLIP models dedicated for discriminative tasks, thereby paving the way for more fine-grained visual applications.

## ACKNOWLEDGMENTS

This work was supported by the grants from the National Natural Science Foundation of China (62525201, 62132001, 62432001) and Beijing Natural Science Foundation (L247006, L257005). This work was partially supported by PKU Kunpeng&Ascend Center of Excellence.

## REPRODUCIBILITY STATEMENT

The main implementations of our proposed models are in Section 4.2 and 4.3. The evaluation metrics is presented in Section 5.1. The prompts for evaluation and implementation details are in Appendix B and C, respectively.

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

## A  QUALITATIVE RESULTS OF VISUAL CONCEPTS

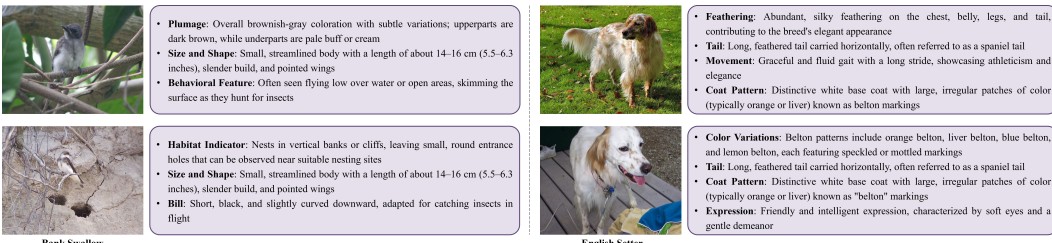

Figure 5: Different image-level visual concepts for objects with the same subcategory.

## B  PROMPT DESIGN

Table 5: Prompt template for FGVR CoT data construction.

---

This is a picture of a {label} with the following visual features: {concepts}. Based on the information provided, please answer the following question. Question: {question}. Note that you MUST first analyze the visual features that help you provide at most four candidate subcategories of the same super-category, then pay attention to the differences between candidate subcategories and make a detailed comparison between them to find evidence that help you make a prediction. The visual analysis process, candidate subcategories, comparison process, and final predicted subcategory are enclosed with <analysis></analysis>, <options></options>, <comparison></comparison>, and <prediction></prediction> tags, respectively, i.e., <analysis>visual analysis process here </analysis><options>candidate subcategories here </options><comparison>comparison process here </comparison><prediction>predicted subcategory here </prediction>.

---

For CLIP models, only closed-world evaluation is conducted. In concrete, CLIP models select the subcategory with the highest cosine similarity to the image feature from the four candidates. We do not include prompt ensembling to fairly compare with MLLMs. For MLLMs, we evaluate FGVR in both closed-world and open-world settings. Example prompts for Fine-R1 are:

(1) Closed-world: "*Given the question: {Question}. This is a fine-grained question, so you need to output fine-grained categories, such as specific animal species or car, airplane model. Output the thinking process in <think></think> and final answer in <answer></answer> tags. The response format should be as follows: <think>...</think><answer>your answer</answer>. Please follow this format exactly.*"

(2) Open-world: "*Given the question: {Question}, based on the options provided in {Options}, output the thinking process in <think></think> and final choice in <answer></answer> tags. The response format should be as follows: <think>...</think><answer>choice</answer>. Please follow this format exactly.*"

## C  IMPLEMENTATION DETAILS

For the CoT SFT data preparation, we utilize the advanced MLLM Qwen2.5-VL-32B (Bai et al., 2025). We adopt Qwen2.5-VL-3B-Instruct and Qwen2.5-VL-7B-Instruct (Bai et al., 2025) as the base models, and full fine-tune them for 10 epochs using Llama-Factory framework (Zheng et al., 2024). After CoT SFT, we subsequently train 3B and 7B models for 10 and 5 epochs on the separate subset of 4-shot training data, using the proposed TAPO instantiated from DAPO baseline (Yu et al., 2025) with clipping factors set to $\epsilon_l = 0.2$, $\epsilon_h = 0.28$, reference KL removed, token-level loss averaging enabled, and dynamic sampling with a maximum of 20 retries. For other RL-related hyperparameters, we adopt the default settings from PAPO (Wang et al., 2025): a global batch size

of 128, a rollout batch size of 384, a learning rate of 1e-6 and weight decay of 1e-2. We use and generate $n = 5$ response per prompt. All training is conducted on 4 A6000 GPUs.

## D    EVALUATED MODELS

Several models are evaluated for comparison, including:

- **CLIP Models:** CLIP-ViT-L/14-336px (shortened as CLIP-L, same below) (Radford et al., 2021), EVA-ViT-G/14 (EVA-G) (Sun et al., 2023), SigLIP (SigLIP-L) (Zhai et al., 2023), and SigLIP2 (SigLIP2-L) (Tschannen et al., 2025).

- **General MLLMs:** Idefics2-8B (Laurençon et al., 2024b), Idefics3-LLaMA3-8B (Laurençon et al., 2024a), LLaVA-v1.6-mistral-7B (Liu et al., 2024b), LLaVA-Onevision-7B (Li et al., 2024), InterVL2.5-2B/4B/8B (Chen et al., 2024), Qwen2-VL-2B/7B (Wang et al., 2024), and Qwen2.5-VL-3B/7B (Bai et al., 2025).

- **Reasoning MLLMs:** DeepPerception-7B (Ma et al., 2025).

Notably, CLIP-L, EVA-G and SigLIP-L are utilized by the LLaVA series (Liu et al., 2024b), the BLIP series (Li et al., 2023b), and Idefics series (Laurençon et al., 2024b) as vision encoders, respectively. Therefore, the MLLMs should theoretically have the competitive or even better FGVR capacity as these vision models.

## E    OPEN-WORLD EVALUATION RESULTS WITH TEXT INCLUSION

Table 6: Open-world FGVR evaluations in terms of text inclusion (%). All results are averaged with 3 trials.

| Models | Seen Categories | | | | | | | Unseen Categories | | | | | | | Avg. |
|---|---|---|---|---|---|---|---|---|---|---|---|---|---|---|---|
| | Air. | Bird | Car | Dog | Flower | Pet | Avg. | Air. | Bird | Car | Dog | Flower | Pet | Avg. | |
| **General MLLMs** | | | | | | | | | | | | | | | |
| Idefics2-8B | 7.49 | 11.02 | 17.94 | 18.01 | 54.27 | 11.12 | 19.98 | 6.76 | 6.01 | 13.37 | 12.14 | 0.38 | 11.45 | 8.35 | 14.16 |
| Idefics3-LLaMA3-8B | 3.30 | 4.79 | 5.57 | 17.04 | 32.37 | 3.72 | 11.13 | 3.16 | 3.50 | 2.97 | 9.59 | 2.12 | 7.57 | 4.82 | 7.98 |
| LLaVA-v1.6-mistral-7B | 2.00 | 3.27 | 9.85 | 16.05 | 24.75 | 13.97 | 11.65 | 2.03 | 3.25 | 8.39 | 9.59 | 0.09 | 7.97 | 5.22 | 8.44 |
| LLaVA-Onevision-7B | 6.44 | 9.27 | 21.33 | 22.55 | 46.50 | 3.26 | 18.23 | 3.46 | 5.54 | 15.36 | 13.01 | 0.14 | 2.68 | 6.70 | 12.46 |
| InternVL2.5-2B | 3.90 | 6.03 | 10.37 | 17.13 | 25.79 | 20.50 | 13.95 | 2.33 | 4.20 | 7.24 | 9.41 | 1.27 | 13.06 | 6.25 | 10.10 |
| InternVL2.5-4B | 8.59 | 7.67 | 16.98 | 19.92 | 25.12 | 16.91 | 15.87 | 7.06 | 7.10 | 10.96 | 11.20 | 1.84 | 10.11 | 8.05 | 11.96 |
| InternVL2.5-8B | 10.04 | 10.97 | 12.78 | 18.76 | 26.34 | 23.12 | 17.00 | 8.64 | 9.00 | 8.42 | 11.53 | 1.79 | 13.66 | 8.84 | 12.92 |
| Qwen2-VL-2B | 34.37 | 25.04 | 60.42 | 41.32 | 59.06 | 4.50 | 37.45 | 42.90 | 8.91 | 42.17 | 30.42 | 3.49 | 4.89 | 22.13 | 29.79 |
| Qwen2-VL-7B | 45.50 | 37.93 | 66.16 | 53.32 | 69.81 | 27.48 | 50.03 | 51.16 | 17.52 | 45.60 | 41.71 | 2.26 | 15.41 | 28.94 | 39.49 |
| Qwen2.5-VL-3B | 37.96 | 48.78 | 58.08 | 51.19 | 61.15 | 13.92 | 45.18 | 40.80 | 22.33 | 43.78 | 36.62 | 5.61 | 12.12 | 26.88 | 36.03 |
| Qwen2.5-VL-7B | 46.85 | 58.28 | 67.14 | 68.90 | 73.09 | 33.55 | 57.97 | 44.78 | 28.60 | 45.88 | 49.03 | 10.80 | 27.19 | 34.38 | 46.17 |
| **Reasoning MLLMs** | | | | | | | | | | | | | | | |
| DeepPerception-7B | 40.66 | 47.63 | 66.62 | 64.78 | 75.47 | 67.37 | 60.42 | 42.37 | 21.33 | 47.28 | 48.21 | 5.04 | 40.12 | 34.06 | 47.24 |
| **Fine-R1-3B (ours)** | 47.30 | 66.09 | 71.15 | 73.45 | 77.16 | 82.58 | 69.62 | 37.27 | **30.68** | 45.73 | 49.79 | **16.50** | 61.02 | 40.17 | 54.90 |
| **Fine-R1-7B (ours)** | **63.44** | **75.22** | **78.88** | **78.41** | **86.92** | **86.76** | **78.27** | **44.85** | 29.77 | **51.52** | **51.24** | 10.37 | **69.86** | **42.94** | **60.61** |

## F    EXPERIMENTS ON MORE BASE MODELS

To further assess generalizability beyond the Qwen series, we apply CoT SFT and TAPO to another base model, openPangu-VL-7B (Luo et al., 2025b). As shown in Table 7, our training pipeline consistently improves performance, demonstrating its effectiveness across different base models.

Table 7: Closed-world FGVR evaluations in terms of accuracy (%) on openPangu-VL-7B (Luo et al., 2025b). All results are averaged with 3 trials.

| Models | Seen Categories | | | | | | | Unseen Categories | | | | | | | Avg. |
|---|---|---|---|---|---|---|---|---|---|---|---|---|---|---|---|
| | Air. | Bird | Car | Dog | Flower | Pet | Avg. | Air. | Bird | Car | Dog | Flower | Pet | Avg. | |
| Zero-shot | 65.33 | 70.60 | 79.21 | 70.09 | 89.47 | 86.49 | 76.87 | 64.61 | 57.94 | 76.19 | 59.53 | 52.85 | 89.48 | 66.77 | 71.82 |
| CoT SFT | 69.13 | 84.98 | 92.18 | **83.11** | 91.71 | **93.89** | 85.83 | **69.72** | **81.35** | **85.73** | 79.36 | **71.10** | 91.29 | **79.76** | 82.80 |
| **CoT SFT + TAPO (ours)** | **70.73** | **85.10** | **92.56** | 82.90 | **92.45** | 93.80 | **86.26** | 68.75 | 80.31 | 85.60 | **80.30** | 70.49 | **91.76** | 79.54 | **82.90** |

## G  GENERAL CAPABILITY

To comprehensively assess the model's general capabilities endowed with FGVR capability, we conduct evaluations on two set of datasets: (1) classification-based VQA benchmark: ImageWikiQA (Zhang et al., 2024e), which is a multiple-choice question-answering dataset collected by feeding the Wikipedia pages of ImageNet classes to GPT-4. (2) General VQA benchmarks: MME (Fu et al., 2023a), MMBench (Liu et al., 2024d), and SEED-Bench (Li et al., 2023a). As shown in Table 8, we find that current MLLMs perform poorly in answering these questions, suggesting that their poor FGVR performance is a fundamental limitation for more advanced capabilities. However, Fine-R1 raises the performance from 54.85% to 58.45%, demonstrating that FGVR is indeed a foundation for MLLMs' advanced capabilities. Moreover, Fine-R1 demonstrates competitive in general-purpose performance and even achieves improvements on MMBench and SEED-Bench. It is worth noting that Fine-R1 is post-trained solely on the FGVR task without incorporating general instruction tuning data, proving that RL drives performance gains not by mere answer memorization. These results suggest that Fine-R1 can serve both as a specialized assistant for users interested in FGVR and as a general-purpose MLLM for broader applications.

Table 8: Performance comparison on three general MLLM benchmarks.

| Model | ImageWikiQA | MME | MMBench | SEED-Bench |
|---|---|---|---|---|
| Qwen2.5-VL-3B | 54.85 | **1591** | 79.30 | 73.74 |
| **Fine-R1-3B (ours)** | **58.45** | 1587 | **79.55** | **73.99** |

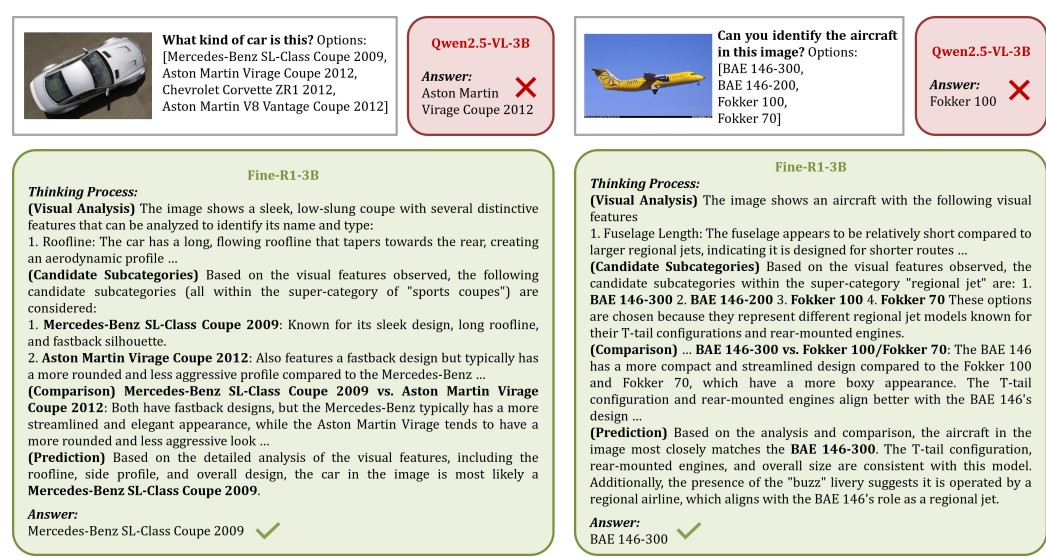

Figure 6: Case study comparing Fine-R1-3B and Qwen2.5-VL-3B on Stanford Car-196 (**Left**) and FGVC-aircraft (**Right**).

## H    QUALITATIVE RESULTS

We provide a qualitative analysis to better demonstrate the effectiveness of our approach. As shown in Figure 6, we can easily observe the model's capability to generate accurate answers through a structured "visual analysis-candidate subcategories-comparison-prediction" process that systematically integrates domain-specific knowledge with visual observations, in contrast to the tendency of the baseline model (i.e., Qwen2.5-VL-3B) to produce incorrect responses directly from superficial pattern recognition.

## I    THE USE OF LARGE LANGUAGE MODELS

LLMs were used solely for polishing writing and error correction in the preparation of this paper, and all suggestions generated by the models were carefully reviewed and verified by the authors.

