# OpenReview forum: "Fine-R1: Make Multi-modal LLMs Excel in Fine-Grained Visual Recognition by Chain-of-Thought Reasoning"
_ICLR.cc/2026/Conference — ICLR 2026 Poster_

### Official Review · Reviewer_ENRW · 2025-10-27

**Soundness:** 2
**Presentation:** 3
**Contribution:** 2
**Rating:** 4
**Confidence:** 5

**Summary:**

This paper introduces Fine-R1, a multi-modal large model (MLLM) designed to excel at fine-grained visual recognition (FGVR). The authors propose a two-stage framework, starting with Chain-of-Thought Supervised Fine-tuning (CoT SFT) to teach the model structured reasoning, followed by Triplet Augmented Policy Optimization (TAPO). TAPO is an RL algorithm based on DAPO that uses anchor, positive, and negative image triplets to handle the FGVR challenges of high intra-class and low inter-class variance.

**Strengths:**

The paper's primary strength lies in its impressive empirical results. Fine-R1 achieves state-of-the-art performance on six FGVR datasets, outperforming general MLLMs, reasoning-focused MLLMs, and even strong contrastive CLIP models. The model shows particularly strong generalization to unseen categories, which is a key challenge in FGVR. The analysis (hypotheses H1-H3) provides an insightful conclusion that the gains come from an improved ability to deploy existing knowledge, rather than learning new features or knowledge.

**Weaknesses:**

Weakness
* Limited Novelty of TAPO: The core algorithmic contribution, TAPO, does not appear to be a novel RL algorithm. It feels like a forced "splicing" of positive ($x^{pos}$) and negative ($x^{neg}$) sampling techniques onto an existing baseline (DAPO/GRPO). The use of a $D_{\text{KL}}$ loss for the $x^{neg}$ sample is a common regularization technique in standard (non-RL) fine-grained classification, which calls into question its novelty as a policy optimization method.
* Unclear Ablation: The paper fails to clearly disentangle the individual contributions of the $x^{pos}$ and $x^{neg}$ components. It is unclear if the $x^{pos}$ (hybrid rollouts) provides any significant benefit on its own. The ablation study is missing crucial comparisons (e.g., Baseline + $x^{pos}$ only, Baseline + $x^{neg}$ only) and also lacks an analysis of the $n_1:n_2$ ratio (anchor vs. positive rollouts).

**Questions:**

If the authors can clearly address the following points during the rebuttal, I am open to reconsidering my score:
1. Can you further justify the novelty of TAPO as an RL algorithm, distinguishing it from simply applying a known FGC regularizer ($x^{neg}$) and a data augmentation strategy ($x^{pos}$) to a DAPO baseline?
2. Can you provide a decoupled ablation study that shows the individual performance contributions of $x^{pos}$ (Intra-class Augmentation) and $x^{neg}$ (Inter-class Augmentation)? I am particularly interested in the (Baseline + $x^{pos}$ only) result.
3. Please provide an ablation study on the ratio of anchor-to-positive rollouts ($n_1$ vs $n_2$).

**Details Of Ethics Concerns:**

no concerns

---

> ### Author Response · Authors · 2025-11-29
> **Rebuttal by Authors**
>
> Thank you for your supportive review and suggestions. Below we respond to the comments in **Weaknesses (W)** and **Questions (Q)**.
>
> > ***W1&Q1: The novelty of TAPO as a policy optimization method.***
>
>
> Thank you for emphasizing the importance of reclaiming the novelty of TAPO as a policy optimization method for FGVR. We recognize that positive and negative sampling techniques are well-established in standard (non-RL, non-MLLM) FGVR task. Nevertheless, we would like to further clarify what makes our approach novel:
>
> 1. **Bridging contrastive and policy-optimization paradigms**. While policy optimization has shown great potential in image classification task (like [a] and [b]) for MLLMs, how it can be optimized for solving the key challenge of fine-grained image classification task hasn't been fully investigated. Contrastive learning with positive and negative samples has shown good performance in FGVR task for CLIP-like models, by bringing features of the same class closer while pushing features of different classes farther apart.  It motivates us to equip policy optimization with contrastive paradigms to make the model more robust to the high intra-class variance and discriminative to the low inter-class variance. **To the best of our knowledge, our Fine-R1 is the first MLLM to surpass various strong CLIP-like models (e.g., SigLIP-L) in FGVR.**
> 2. **Modeling intra-class variance by rollout-level augmentation**. We collect trajectories from both $x$ and $x^{pos}$, but during policy optimization, we treat all rollouts as if they originate from $x$, regardless of their actual source.  In this way, we facilitate the model to focus on the distinguishing visual cues of the class while ignoring the variations in certain visual features among different individuals of the same class, as well as category-irrelevant factors such as background, lighting, and viewpoint.
> 3. **Modeling inter-class variance by distribution contrast.** *Vector embeddings* of anchor and negative samples are typically contrasted for CLIP-like models that use representation similarity to do FGVR. For generative MLLMs, we instead contrast *KL divergence between the output distribution* conditioned on the anchor image and negative image.
>
> > ***W2&Q2&Q3: Ablation on the individual performance contributions of $x^{pos}$ and $x^{neg}$; ablation on the $n_1:n_2$ ratio.***
>
> Thank you for pointing out these two important ablation studies. We additionally conduct experiments evaluating the individual contributions of $x^{pos}$ and $x^{neg}$, as well as positive-to-anchor ratio $n_1:n_2$:
>
> 1. **$x^{pos}$ and $x^{neg}$ are both effective in bringing performance contributions.** Both $x^{pos}$ and $x^{neg}$ outperform baseline (i.e., CoT SFT + DAPO, as shown in the first row), and combining them together achieves the best results, confirming that Fine-R1 benefits from complementary augmentations (i.e., intra-class and inter-class augmentation).
>
>     | $x^{pos}$ | $x^{neg}$ | Seen Avg. | Unseen Avg. | Total Avg. |
>     | :-------- | --------- | :-------- | :---------- | :--------- |
>     |           |           | 71.67     | 59.95       | 65.81      |
>     | ✓         |           | 72.35     | 60.58       | 66.47      |
>     |           | ✓         | 72.28     | 60.29       | 66.29      |
>     | ✓         | ✓         | **73.35** | **61.29**   | **67.32**  |
>
> 2. **Nearly equal rollouts from $x^{anchor}$ and $x^{pos}$ yields more gains.** We control $n_1+n_2=10$, and change the anchor-to-positive rollout ratio $n_1:n_2$. We test in open-world FGVR in terms of $SS_{relative}$. $n_1:n_2=1$ achieves the best performance, further confirming that the performance gain of intra-class augmentation is from increasing the diversity of rollouts instead of generating more rollouts (i.e., using $x$ merely to generate more rollouts, as shown in the first row).
>
>     | $n_1$ | $n_2$ | Air.      | Bird      | Car       | Dog       | Flower    | Pet       | Avg.       |
>     | :---- | ----- | :-------- | :-------- | :-------- | --------- | --------- | --------- | :-------- |
>     | 10    | 0     | 42.81     | 56.09     | 73.00     | 69.42     | 37.71     | 77.81     | 59.47     |
>     | 8     | 2     | 42.00     | 56.13     | 72.27     | 69.06     | 37.45     | 78.36     | 59.21     |
>     | 5     | 5     | **46.43** | **58.00** | **74.60** | **70.08** | **39.54** | **79.11** | **61.29** |
>     | 2     | 8     | 41.33     | 56.28     | 72.30     | 69.50     | 37.84     | 78.71     | 59.33     |
>     | 0     | 10    | 42.78     | 56.53     | 72.43     | 69.20     | 36.97     | 78.42     | 59.39     |
>
> [a] Visual-RFT: Visual Reinforcement Fine-Tuning, ICCV 2025.
>
> [b] DeepPerception: Advancing R1-like Cognitive Visual Perception in MLLMs for Knowledge-Intensive Visual Grounding, Arxiv 2025.

---

### Official Review · Reviewer_8sko · 2025-10-29

**Soundness:** 3
**Presentation:** 3
**Contribution:** 3
**Rating:** 6
**Confidence:** 4

**Summary:**

The paper proposes Fine-R1, a multimodal large language model (MLLM) framework for fine-grained visual recognition (FGVR). It introduces a two-stage R1-style training pipeline: (1) Chain-of-Thought Supervised Fine-Tuning (CoT-SFT), where the model learns structured reasoning steps from synthesized CoT data; and (2) Triplet Augmented Policy Optimization (TAPO), a reinforcement-learning method that augments intra- and inter-class examples (anchor, positive, negative) to improve robustness and discrimination. Fine-R1 achieves superior accuracy on six FGVR benchmarks, outperforming both general and reasoning MLLMs (e.g., Qwen2.5-VL, DeepPerception) and even contrastive CLIP models, especially in few-shot and unseen-category settings.

**Strengths:**

**Clear motivation**: The paper clearly articulates the limitations of existing MLLMs in fine-grained visual recognition and motivates the need for improved reasoning and generalization capabilities.

**Methodological novelty**: The proposed TAPO combines reinforcement learning with triplet-based augmentation, conceptually bridging contrastive and policy-optimization paradigms.

**Strong empirical results**: Fine-R1 consistently surpasses prior MLLMs and CLIP baselines across multiple FGVR datasets and evaluation settings (closed/open-world, seen/unseen).

**Well-structured analyses**: The ablation studies and hypothesis testing (H1–H3) are thoughtful, showing that Fine-R1’s gains stem from better knowledge deployment rather than merely improved visual features or memorization.

**Weaknesses:**

**Limited data scale for CoT-SFT**: The CoT dataset reportedly contains only 404 samples, raising doubts about generalization and potential overfitting to synthetic patterns.

**Evaluation bias toward Qwen-based baselines**: All base models are Qwen-VL variants; cross-model validation (e.g., on LLaVA or InternVL foundations) is missing, which might limit claims of generality.

**Complexity vs. gain**: TAPO adds considerable training and sampling overhead (triplet construction, multi-rollout reward computation), but the improvement over DAPO (+1.6%) is relatively modest.

**Conceptual overlap**: While well-positioned as an R1-style method, the framework’s connection to previous RL-based reasoning systems (e.g., Visual-RFT, VLM-R1) could be made more precise to clarify incremental novelty.

**Interpretability of CoT generation**: The reasoning chains are auto-synthesized by another MLLM (Qwen2.5-VL-32B), but no human verification or quality metrics are provided, leaving uncertainty about rationale faithfulness.

**Questions:**

**1. Data efficiency and generalization:**
The CoT-SFT dataset contains only 404 samples. Could the authors elaborate on how they ensure generalization beyond this limited synthetic set? For instance, were any experiments conducted to test scaling behavior when using larger or more diverse CoT data?

**2. On cross-model validation:**
Since all base models are Qwen-VL variants, have the authors attempted to reproduce the results on alternative architectures (e.g., LLaVA or InternVL) to confirm that the proposed TAPO framework generalizes across backbones?

**3. On training efficiency and computational cost:**
TAPO introduces triplet sampling and additional rollouts. Could the authors quantify the training-time or GPU-hour overhead compared to DAPO or standard GRPO?

**4. On incremental novelty and relation to prior work:**
The paper positions Fine-R1 as an R1-style framework. Could the authors clarify the specific conceptual or algorithmic distinctions from prior RL-based reasoning methods such as Visual-RFT, Vision-R1, or VLM-R1? What unique design choices make TAPO fundamentally different rather than a variant?

**5. On CoT faithfulness and quality control:**
The CoT rationales are generated automatically by Qwen2.5-VL-32B. Did the authors evaluate their correctness or consistency? How sensitive is model performance to potential noise in these synthesized CoTs?

I will adjust my score based on the authors’ response.

---

> ### Author Response · Authors · 2025-11-29
> **Rebuttal by Authors**
>
> Thank you for your supportive review and suggestions. Below we respond to the comments in **Weaknesses (W)** and **Questions (Q)**.
>
> > ***W1&Q1: Data efficiency and generalization.***
>
> Thank you for emphasizing the importance of testing scaling behavior when using larger CoT data. We additionally construct more CoT data from the same model Qwen2.5-VL-32B, and evaluate $SS_{relative}$ on unseen categories.
>
> | #CoT | Air.      | Bird      | Car       | Dog       | Flower    | Pet       | Avg.       |
> | :--- | :-------- | :-------- | :-------- | --------- | --------- | --------- | :-------- |
> | 404  | 42.70     | 55.54     | 72.91     | 67.20     | 37.09     | 76.53     | 58.66     |
> | 804  | 52.38     | 56.13     | 79.21     | **68.33** | **38.30** | 77.99     | 62.06     |
> | 1199 | **54.90** | **56.66** | **79.28** | 67.40     | 37.98     | **79.63** | **62.64** |
>
> As shown in the table, the model performance increases with the number of CoT data, confirming that the model *does not overfit to synthetic patterns in the limited synthetic set*. Additionally, we can observe that quality out-weights quantity of CoT data, alleviating the cost to construct a large scale of data for CoT SFT, showing the high data efficiency.
>
> > ***W2&Q2: On cross-model validation.***
>
> We appreciate the reviewer’s concern. In this work, we primarily focus on the Qwen2.5-VL family because it is currently the most widely used and infrastructure-supported model series. It also serves as the default backbone in most recent multimodal RLVR studies [a,b,c,d,e]. As a result, we believe that conclusions drawn from Qwen2.5-VL experiments are both robust and well aligned with prevailing community practice.
>
> Nevertheless, we agree that evaluating additional model families can further strengthen the evidence of the effectiveness. To this end, we have tried to apply TAPO to Qwen2-VL and InternVL3.5 models:.
>
> 1. **New results on Qwen2-VL**: We conduct experiments on the Qwen2-VL-2B model which is utilized as the base model in the first work to apply GRPO for image classification [f]. Architecturally, Qwen2.5-VL differs from Qwen2-VL through the use of an updated vision encoder and language model and a new vision–language fusion module.
>
>     | Method              | Air.      | Bird      | Car       | Dog       | Flower    | Pet       | Avg.       |
>     | :------------------ | :-------- | :-------- | :-------- | --------- | --------- | --------- | :-------- |
>     | Zero-shot           | 42.91     | 45.68     | 54.93     | 63.71     | 34.51     | 51.30     | 48.84     |
>     | CoT SFT+DAPO        | 52.29     | 55.26     | **80.42** | 68.68     | 40.60     | 76.68     | 62.32     |
>     | CoT SFT+TAPO (ours) | **61.07** | **56.85** | 78.17     | **71.19** | **41.62** | **79.01** | **64.65** |
>
> As shown in the table, the consistent gains compared to CoT SFT+DAPO again shows the generality of our proposed TAPO to different model architectures.
>
> 2. **InternVL3.5 attempt**: We also explored integrating the InternVL3.5 series into our codebase, even though it is not yet officially supported by the EasyR1 framework. Unfortunately, this led to kernel-level errors that we are unlikely to fully resolve within the rebuttal period. We will continue coordinating with the EasyR1 and vLLM maintainers to further diagnose and address these issues.
>
> > ***W3&Q3: On training efficiency and computational cost.***
>
> We quantify the training-time overhead on 8 A6000 GPUs. As our method involves relatively simple augmentation steps without complex training protocols, and we pre-construct triplets before training, the increase in training time per step is reasonable, with ours taking 26.3 minutes compared to DAPO's 21.1 minutes approximately.

---

> ### Author Response · Authors · 2025-11-29
> **Rebuttal by Authors**
>
> > ***W4&Q4: On incremental novelty and relation to prior work.***
>
> Thank you for emphasizing the importance of discussing prior RL-based reasoning methods. We recognize that RL has shown great success in MLLM reasoning (like Visual-RFT, Vision-R1, and VLM-R1). We have included foundational works in our revised related work section.
>
> Nevertheless, we would like to further clarify what makes our approach novel:
>
> 1. **Bridging contrastive and policy-optimization paradigms**. While policy optimization has shown great potential in image classification task (like [a] and [b]) for MLLMs, how it can be optimized for solving the key challenge of fine-grained image classification task hasn't been fully investigated. Contrastive learning with positive and negative samples has shown good performance in FGVR task for CLIP-like models, by bringing features of the same class closer while pushing features of different classes farther apart.  It motivates us to equip policy optimization with contrastive paradigms to make the model more robust to the high intra-class variance and discriminative to the low inter-class variance. **To the best of our knowledge, our Fine-R1 is the first MLLM to surpass various strong CLIP-like models (e.g., SigLIP-L) in FGVR.**
> 2. **Modeling intra-class variance by rollout-level augmentation**. We collect trajectories from both $x$ and $x^{pos}$, but during policy optimization, we treat all rollouts as if they originate from $x$, regardless of their actual source.  In this way, we facilitate the model to focus on the distinguishing visual cues of the class while ignoring the variations in certain visual features among different individuals of the same class, as well as category-irrelevant factors such as background, lighting, and viewpoint.
> 3. **Modeling inter-class variance by distribution contrast.** *Vector embeddings* of anchor and negative samples are typically contrasted for CLIP-like models that use representation similarity to do FGVR. For generative MLLMs, we instead contrast *KL divergence between the output distribution* conditioned on the anchor image and negative image.
>
> > ***W5&Q5: On CoT faithfulness and quality control.***
>
> Thank you for emphasizing the importance of discussing rationale faithfulness. In fact, **the correctness and consistency of our synthesized CoT rationales are strictly ensured** by human verification:
>
> 1. We sample multiple responses until the CoT leads to exactly matched subcategory.
> 2. We detect CoT with mixed language and manually correct them to English.
> 3. We manually check the predicted subcategory within <prediction></prediction> in the CoT rationales, and maintain the samples whose predictions are both *included in the candidate subcategories* within <candidates></candidates> and *consistent with the ground truth* within <answer></answer>.
>
> To further assess the model’s sensitivity to potential noise in synthesized CoTs, we also generate CoTs using Qwen2.5-VL-7B instead of Qwen2.5-VL-32B, keeping the quantity fixed but intentionally lowering their quality. This substitution results in a drop in SFT accuracy (59.81% vs. 64.21%), reinforcing the importance of our quality-control procedures for CoT SFT.
>
>
>
> [a] R1-onevision: Advancing generalized multimodal reasoning through cross-modal formalization, Arxiv 2025.
>
> [b] Noisyrollout: Reinforcing visual reasoning with data augmentation, NeurIPS 2025.
>
> [c] Vl-rethinker: Incentivizing self-reflection of vision-language models with reinforcement learning, Arxiv 2025.
>
> [d] Visionary-r1: Mitigating shortcuts in visual reasoning with reinforcement learning, Arxiv 2025.
>
> [e] R1-ShareVL: Incentivizing Reasoning Capability of Multimodal Large Language Models via Share-GRPO, Arxiv 2025.
>
> [f] Visual-RFT: Visual Reinforcement Fine-Tuning, ICCV 2025.

---

### Official Review · Reviewer_Ace4 · 2025-10-31

**Soundness:** 2
**Presentation:** 2
**Contribution:** 2
**Rating:** 4
**Confidence:** 2

**Summary:**

This paper targets FGVR by proposing a two-stage approach to improve MLLMs. The first stage, CoT SFT, uses supervised fine-tuning with a structured Chain-of-Thought to teach the model an interpretable, fine-grained reasoning process. The second stage, TAPO, employs a triplet-augmented policy optimization to sharpen the model's ability to distinguish between highly similar classes. In a 4-shot, base-to-new setting across six FGVR datasets, the method reports significant gains over both general-purpose MLLMs and strong contrastive models like SigLIP.

**Strengths:**

1、t's widely acknowledged that general MLLMs underperform contrastive models like CLIP/SigLIP on fine-grained tasks. The paper's attempt to close this gap using structured CoT and reinforcement learning, rather than relying solely on massive labeled datasets, is a practical and appealing direction.

2、The two-stage approach is well-motivated. The CoT SFT stage provides an interpretable reasoning framework ("visual analysis → candidate subclasses → comparison → prediction"), while the triplet-based policy optimization (TAPO) directly targets the core challenge of FGVR: maximizing inter-class variance while minimizing intra-class variance.

3、The experiments are extensive, covering both closed-set and open-set scenarios. The use of multiple evaluation metrics (including semantic similarity) and thorough ablations helps to clearly identify the sources of performance improvement.
Weaknesses & Suggestions

**Weaknesses:**

1、The method feels like an application of existing CoT and RL techniques, not a new paradigm for FGVR. A direct comparison against a generic CoT prompt is needed to prove the proposed reasoning structure is truly beneficial.

2、Using a SigLIP encoder to calculate a key metric while also comparing against SigLIP is a potential conflict. The results should be cross-verified with another encoder (like CLIP's) to ensure fairness.

3、The reported gains are marginal and lack error bars, making them unconvincing given the high variance of RL and CoT methods.

4、It's unclear if the gains come from the reasoning structure or just from generating longer text, a known confounder. The paper needs length-controlled experiments to prove its central claim. Weak Baselines.

5、The CLIP/SigLIP baselines seem under-tuned, as they lack standard optimizations like prompt ensembling.

**Questions:**

1、Novelty of CoT Application Needs Clarification. The core idea is to combine structured CoT with policy optimization. However, using CoT to enhance reasoning is already a well-explored area in LLMs. The paper needs to more clearly articulate the fundamental difference between its "CoT SFT" and existing work on few-shot CoT prompting or standard CoT-based supervised fine-tuning.

2、Details on Semantic Similarity Metric are Lacking. The paper relies on a SigLIP text encoder to calculate semantic similarity, which is a key metric. However, it needs to provide more details on the threshold selection, the metric's sensitivity to different class granularities, and the potential impact of text normalization or synonyms.

3、Clarity on Acronyms. Several new acronyms ("CoT SFT," "No-Thinking-RL," "TAPO," etc.) should be defined with their full names upon first use to improve readability.

---

> ### Author Response · Authors · 2025-11-29
> **Rebuttal by Authors**
>
> Thank you for your supportive review and suggestions. Below we respond to the comments in **Weaknesses (W)** and **Questions (Q)**.
>
> > ***W1&Q1: Novelty of CoT SFT; comparison against generic CoT prompt.***
>
> Thank you for emphasizing the importance of discussing existing works on few-shot CoT prompting and standard CoT-based supervised fine-tuning (CoT SFT). We recognize that using CoT to enhance reasoning is well-established in LLMs and MLLMs. We additionally include foundational works in our revised related work section *CoT Reasoning with MLLMs*.
>
> Our stage1 utilizes CoT SFT instead of CoT prompting, since prompt variation has a limited impact on the image classification performance of  MLLMs [a]. We would like to further clarify what makes our approach novel compared to generic CoT SFT:
>
> **Candidate subcategories elicits differential reasoning**. The correct label often appears among the K sampled response, yet fails to get it correct as Pass@1, indicating possible over-reliance on coarse, salient attributes shared by related categories and may benefit from a fine-grained, differential reasoning to separate semantically similar categories. Instead of using generic CoT of "*visual analysis and prediction*", we utilize the structured reasoning procedure of “*visual analysis, candidate subcategories, comparison, and final prediction*”, eliciting the model to first propose candidate subcategories (the most likely categories base model confuses it for) and text utilize text knowledge to resolve this confusion by detailed comparison between candidates. The closed-world accuracy drops significantly when using a generic CoT prompt (85.19%→74.79%), confirming that the reasoning structure is truly beneficial.
>
> > ***W2: Cross verification with another encoder.***
>
> Thank you for pointing out this misunderstanding, which is indeed easy to make. We use SigLIP encoder to calculate relative semantic similarity ($SS_relative$) *in the open-world evaluation*. Since CLIP-like models like SigLIP fail to do open-world FGVR,  they are not included in comparison in terms of $SS_relative$. Thus, **there is no potential conflict.**
>
> > ***W3: Performance with error bars.***
>
> Thank you for emphasizing the importance of providing performance with error bars. The reported results in the manuscript are calculated in the setting of *greedy sampling with do_sample=False*, avoiding the high variance of RL and CoT methods. We additionally calculate the metrics with error bars by setting do_sample=True, temperature=1.0, top_k=50, top_p=0.95, num_return_sequences=3. Taking closed-world evaluation on Aircraft as an example, the accuracy of Qwen2.5-VL-3B (+CLS-RL) and ours are 76.06±0.91 and 78.09±0.89, respectively. It shows that **the gains of our method are convincing even using sampling generation with higher randomness.**
>
> > ***W4: It is unclear if the gains come from the reasoning structure or generating longer text.***
>
> We understand the reviewer’s concern about the known confounder. We prompt the Qwen2.5-VL-3B to directly generate longer text using the zero-shot CoT prompting. With the generated text gets longer (6.76 tokens→88.58 tokens on average) , the model achieves only a marginal performance gain (74.68%→74.79%), confirming that **only generating longer text without learning optimized reasoning structure cannot lead to performance gain effectively**.
>
> > ***W5: Prompt ensembling results of CLIP-like models.***
>
> Thank you for suggesting adding standard optimizations like prompt ensembling for CLIP-like models. We did not include prompt ensembling to fairly compare with MLLMs following [a]. Nevertheless, we additionally add prompt ensembling results for SigLIP-L using 80 prompt templates from ImageNet dataset. The prompt ensembling brings no performance gains (84.44%→84.35%), which again confirms the superiority of our Fine-R1 (85.19%) compared to strong CLIP-like models even with optimization.

---

> ### Author Response · Authors · 2025-11-29
> **Rebuttal by Authors**
>
> > ***Q2: Details on semantic similarity metric.***
>
> Thank you for emphasizing the importance of discussing on the details of semantic similarity metric.
>
> 1. **Threshold selection:** $SS_{relative}$ is defined as:
> $$
> SS_{relative}=max(0, \frac{Sim(c,c*)-Sim(\hat{c},c*)}{1-Sim(\hat{c},c*)})\in[0,1]
> $$
> where the lower bound and the upper bound is reached when the prediction matches the super-category and ground truth subcategory, respectively.
>
> 2. **Sensitivity to different class granularities:** Considering that the *absolute* semantic similarity is sensitive to the super-category (e.g., Aircraft, Bird, etc.), we propose *relative* semantic similarity which measures the relative gains compared to predicting the super-category. Taking CUB-200 as example, $SS_{relative}$ when correctly predicting label in the Order level is smaller than the Family level (0.1449 vs 0.1684), confirming that it can illustrate the granularities of the predictions.
>
> 3. **The potential impact of text normalization or synonyms:** Since SigLIP encoder is trained on a large corpus, it is relatively robust to text normalization or synonyms. Taking CUB-200 as example, the similarity between the common name (i.e., used as the label) and scientific name is relatively high (0.6635).
>
> > ***Q3: Clarity on Acronyms.***
>
> Thank you for emphasizing the importance of clarity on acronyms. "CoT SFT" and "TAPO" had been defined with their full names upon first use in Introduction section. We newly add "No-Thinking RL" with their full name ("No-Thinking Reinforcement Learning") in Introduction section.
>
> [a] Why are visually-grounded language models bad at image classification? NeurIPS 2024.

---

### Author Response · Authors · 2025-11-29
**Response Summary**

Dear Area Chairs and Reviewers,

We sincerely thank you for your time, thoughtful comments, and constructive engagement throughout the review process. We greatly appreciate the reviewers' recognition of our work's **clear motivation** (Reviewer Ace4&8sko), **methodological novelty** (Reviewer 8sko), **impressive empirical results** (Reviewer Ace4&8sko&ENRW), and **insightful analyses** (Reviewer Ace4&8sko&ENRW).

In our response, we have addressed all reviewers' questions, clarified potential misunderstandings, and conducted additional experiments to strengthen our findings:

**1. Difference between existing work**

- **To the best of our knowledge, our Fine-R1 is the first MLLM to surpass various strong CLIP-like models (e.g., SigLIP-L) in FGVR**: It's widely acknowledged that general MLLMs underperform contrastive models like CLIP/SigLIP on fine-grained tasks. Our work  bridges this gap and strongly indicates *the potential of generative MLLMs for discriminative vision tasks*.
- **Key distinction from existing CoT SFT approaches**: Instead of generic CoT of “*visual analysis and final prediction*”, our method utilizes the structured reasoning procedure of “*visual analysis, candidate subcategories, comparison, and final prediction*”, additionally eliciting the model to first propose candidate subcategories and then utilize text knowledge to resolve this confusion by detailed comparison between candidates.
- **Key distinction from existing RL approaches**: Instead of standard RL for image classification, our method *bridges contrastive and policy-optimization paradigms*. We model intra-class variance and inter-class variance by rollout-level augmentation and distribution contrast, respectively, providing a promising way for settling the key challenge of FGVR for generative MLLM models.

**2. More ablation and analysis**

* **Decoupled ablation study** shows the complementary benefits of our proposed Intra-class Augmentation and Inter-class Augmentation.
* **Ablation study on the ratio of anchor-to-positive rollouts** confirms that the performance gain of Intra-class Augmentation is from increasing the diversity of rollouts instead of just generating more rollouts, a known confounder.
* **Cross-model validation** shows that the proposed TAPO generalizes across model backbones.
* **Length-controlled experiments** confirm that the gains come from the reasoning structure instead of just from generating longer text.
* **Ablation on the number of CoT samples in stage 1** shows the high data efficiency and generalization of our constructed high-quality synthetic set for CoT SFT.
* **Training-time comparison** shows that our proposed TAPO has little increase in terms of training time per step.
* **Results of SigLIP with prompt ensembling** show the superiority of Fine-R1 compared with CLIP-like models even with optimization.

**3. Clarifications**

* **More details of $SS_{relative}$**: We provide more details of the metrics $SS_{relative}$ on the threshold selection, the metric's sensitivity to different class granularities, and the potential impact of text normalization or synonyms.
* **More details of CoT quality control steps:** We present more details of our human verification steps to ensure the correctness and consistency of synthetic CoTs.

Thank you again for your valuable feedback. **We have updated the manuscript with a new revision.** The revised and newly added contents are highlighted in the blue format in the revised manuscript. We will release code/models to merit further exploration in the community.

---

### Meta-Review · Area_Chair_gUvx · 2026-01-06

**Summary:**

The paper initially received mostly negative scores: 6, 4, 4. The main concerns include: (1) difference to previous works; (2) more ablation and analysis; (3) Clarifications on formulation and experiment. The authors have provided a detailed rebuttal to respond to the reviewers’ concerns. The AC has carefully read the reviews and the rebuttal, and finds that the authors have mostly addressed these concerns.

Specifically, for the concerns of Reviewer Ace4: the authors have addressed the concerns of novelty against generic CoT prompt and more ablation studies.

For the concerns of Reviewer 8sko: the authors have addressed the concerns of experiments with more baselines and novelty against previous works.

For the concerns of Reviewer ENRW: the authors have addressed the concerns of novelty over previous works and critical ablation studies.


Given these considerations, the AC believes the authors have addressed the main concerns, and that the reviewers are likely to change their original scores to all positive scores: 6, 6, 6. The AC thus would like to recommend acceptance to this paper.

**Reviewer Concerns:**

The main concerns of the reviewers were: (1) difference to previous works; (2) more ablation and analysis; (3) clarifications on formulation and experiment. These concerns were mostly solved.

Remaining concern of Reviewer Ace4:  Cross verification with another encoder. The authors did not provide additional results.

Remaining concerns of Reviewer 8sko: Limited data scale for CoT-SFT. The authors have provided results with more samples. However, the scale is still limited (from 404 samples to 1199 samples).

**Reviewer Scores:**

Reviewer Ace4 may change the score from 4 to 6 as the main concerns were solved.

Reviewer 8sko may keep the score of 6 as the main concern were solved but one minor concern was not solved.

Reviewer ENRW may change the score from 4 to 6 as the main concerns were solved.

---

### Decision · Program_Chairs · 2026-01-26

Accept (Poster)